# Addressing the Endometriosis Knowledge Gap for Improved Clinical Care—A Cross-Sectional Pre- and Post-Educational-Intervention Study among Pakistani Women

**DOI:** 10.3390/healthcare11060809

**Published:** 2023-03-09

**Authors:** Muhammad Saad, Aiman Rafiq, Anam Jamil, Zouina Sarfraz, Azza Sarfraz, Karla Robles-Velasco, Ivan Cherrez-Ojeda

**Affiliations:** 1Department of Research, Services Institute of Medical Sciences, Lahore 54000, Pakistan; 2Department of Research, Nishtar Medical University, Multan 66000, Pakistan; 3Department of Research, Continental Medical College, Lahore 54000, Pakistan; 4Department of Research and Publications, Fatima Jinnah Medical University, Lahore 54000, Pakistan; 5Department of Pediatrics and Child Health, The Aga Khan University, Karachi 74800, Pakistan; 6Department of Allergy and Pulmonology, Universidad Espíritu Santo, Samborondón 092301, Ecuador

**Keywords:** endometriosis, knowledge, women’s health, symptoms, awareness, satisfaction

## Abstract

This study explored the baseline knowledge and symptoms associated with endometriosis, educated women about endometriosis, and documented the improvement in endometriosis knowledge. A brief intervention with an educational brochure on endometriosis was administered among 303 female participants. A paired-sample t-test and one-way ANOVA were run to compute endometriosis knowledge scores. In total, 49.5% of the participants had consistent access to annual healthcare, 25.1% were healthcare workers, and 45.2% had an undergraduate education. The entire sample had a pre-test endometriosis knowledge score of 4.2 (SD = 2.48), and the score was 6.3 (SD = 2.3) on the post-test. One-way ANOVA yielded no significant improvement in knowledge scores across the groups with consistent and inconsistent access to annual healthcare, despite a significant overall increase in scores (t = 16.63, df = 302, *p* < 0.001). We identified a significant improvement in the knowledge concerning endometriosis. Educational strategies among women who are menstruating are essential for addressing diagnostic delays and bridging the endometriosis knowledge gap.

## 1. Introduction

Endometriosis is an estrogen-dependent chronic disorder that is defined by endometrial tissue outside the uterine cavity in the form of lesions, and it was first described in 1925 [1]. The disorder is multifactorial and has various etiological factors, including autoimmunity and genetic predisposition, environmental factors, including dioxin and polychlorinated biphenyl, and lifestyle risk factors, such as alcohol and caffeine [2]. The revised American Society of Reproductive Medicine (rASRM) scores define endometriosis as (i) superficial endometriosis (SE), with small peritoneal lesions that are typically present in the pelvis, (ii) ovarian endometrioma (OMA), which is a cyst containing chocolate-colored fluid that occurs due to repeated hemorrhages within the endometriotic tissue during menstruation and is associated with adhesions in the posterior leaf of the broad ligament or pelvic call, and (iii) deep endometriosis (DE), which is associated with an invasion of the peritoneum of greater than 5 mm [3]. All three phenotypes can co-exist among women with DE, and they are present in 48% of diagnosed endometriosis cases [4]. Symptoms experienced by women with endometriosis include pelvic pain associated with urination and defecation during menstruation, pelvic pain in between periods, dyspareunia, and menorrhagia [5,6]. As expected, DE is associated with a more severe clinical presentation than that of SE, as it includes subfertility [7,8]. Given the chronic and potentially severe nature of the symptoms associated with endometriosis, it considerably impacts the quality of life of women [9,10].

The prevalence of endometriosis is estimated to be 5–15% [11]; it is diagnosed in women with dysmenorrhea (40–60%), pelvic pain (71–87%), and subfertility (21–47%) [12]. However, the previous literature demonstrated that these prevalence estimates were based on data from women who were either hospitalized or undergoing surgery; therefore—and due to due to their differences in definition and diagnostic staging—they are not predictive of the general population [13,14,15]. The challenge in diagnosing endometriosis is that the key symptoms, including pain and subfertility, may be associated with various causes. Further, the correlation of symptoms with the severity or extent of the disease is not linear as is defined by the current staging systems, which limits surgical assessment of the disease [16]. Those who receive a diagnosis of endometriosis present with a delay of 4–11 years from the onset of the first symptom [17]. This delay may be due to the “normalization” of symptoms, as well as misdiagnoses experienced by women [18]. As a result, disease progression and adhesion formation may proceed further, thus potentially compromising fertility and increasing the risk of chronic pelvic pain [19]. Soliman et al. demonstrated that the mean time from the first consultation to diagnosis was reduced when women were assessed with nonsurgical clinical methods; however, there is no consistent or standardized algorithm for this [20].

The objective of the following study is to identify the baseline knowledge and symptoms associated with endometriosis in the Pakistani population, as well as to educate women on endometriosis. The knowledge regarding endometriosis was re-assessed following the education of the participants.

## 2. Materials and Methods

A brief intervention with an educational brochure on endometriosis (image-based pre-/post-test design) designed to improve the knowledge about endometriosis among the participants was applied (Figure 1).

The study was conducted across Pakistan from May 2021 to August 2021. The study was conducted by using the Google Forms software. Using platforms including Facebook, WhatsApp, and LinkedIn, 303 females consented to take part in the study, and all of them completed the survey. We screened the participants to assess if they entered with a high or low mean score for knowledge by using an a priori scale, which is presented in Table 1. As it was a special educational intervention that we tested, we used a grouping variable (1 = women who always have access to annual healthcare; 2 = women with inconsistent access, i.e., often/sometimes/rarely/never have access to annual healthcare), and both groups underwent the educational intervention. To be considered eligible for the study, the participants were required to be female and could be in any age group. The study’s protocol was approved by the ethics committee before the surveys were distributed, and a waiver under the Declaration of Helsinki was obtained.

The endometriosis knowledge score (EKS) ranged from high (10) to low (0) across four questions on the pre-test and post-test, in which the same questions (provided in the Appendix A) were used. The difference between the EKS after the post-test and after the pre-test was computed to quantify the post-intervention knowledge difference. The educational intervention was administered to the groups of those who always had access to healthcare (AH) and those with inconsistent access to healthcare (IH). We aimed to determine if the educational intervention was beneficial in the sample by noting the changes in the endometriosis knowledge scores. When the participants took the pre-intervention survey, the pre-test score was considered the dependent variable, and the grouping variable (AH = always had access to healthcare; IH = Inconsistent access to healthcare) was the independent variable.

The effectiveness of the Intervention was measured by using the mean difference in the EKS before and after the intervention. With an equal sample size in both groups, the minimum number of samples required to detect a difference between the pre-and post-intervention mean EKS was 42 participants with a power of 80% at a 95% confidence interval [21].

The data were summarized by using frequencies and percentages. Pre- and post-intervention differences were assessed by using paired-sample t-tests. An independent sample t-test/one-way ANOVA was used to assess knowledge about endometriosis at the baseline. One-way ANOVA was used to assess factors associated with endometriosis knowledge. The significance was set a priori for all comparisons at *p* < 0.05.

## 3. Results

The study’s participants were aged 18–29 years (77.9%), 30–39 years (12.9%), <18 years (4.6%), and >40 years (4.6%). The majority had an undergraduate level of education (45.2%), with 36% being graduates, 9.9% having a middle/intermediate school education, and 8.9% having a high school education. There were 25.1% healthcare workers and 74.9% non-healthcare workers (*p* < 0.001). In total, 150 (49.5%) participants always had access to annual healthcare, whereas 50.5% of the respondents had inconsistent access to annual healthcare. The sample characteristics are listed in Table 2.

The participants’ responses to the questions on endometriosis knowledge pre-and post-intervention are depicted in Figure 2.

Across the entire cohort of women (*n* = 303), the independent variables (AH = always had access to healthcare; IH = inconsistent access to healthcare) were associated with all other variables; both the inferential and descriptive statistical findings are tabulated in Table 3. Before the educational intervention was administered, 49.2% of the participants were somewhat or completely satisfied with their knowledge of symptoms related to periods (*p* = 0.008). However, 18.2% had not had pelvic pain with periods in the last three months, 29% always had pelvic pain with periods, 22.4% often had pelvic pain with periods, and 30.4% occasionally had pelvic pain with periods in the last three months. Concerning the question of whether the participants had had pelvic pain between one period cycle and the next in the last three months, 7.9% always had pelvic pain, 13.5% often had pain, 31.4% occasionally had pelvic pain, and 47.2% never had pelvic pain. When the participants were questioned about pelvic pain while urinating or defecating during periods in the last three months, 64% of the respondents reported that they never had pain, whereas 6.9% always had pain, 7.3% often presented with pelvic pain, and 21.8% occasionally had pelvic pain while urinating or defecating (*p* = 0.02) (Table 3).

When the participants were asked about concerns related to period symptoms, 28.7% were extremely or moderately concerned, 41% were slightly or somewhat concerned, and 30.4% were not at all concerned (*p* < 0.001). Of all participants, only 10.6% suspected that symptoms related to periods were due to endometriosis, 20.5% did not suspect endometriosis as a probable cause but considered it to be related to periods, and the majority (69%) did not suspect or consider endometriosis to be related to period symptoms (*p* < 0.001). Of the 31 participants that suspected endometriosis to be a cause of period symptoms, 15 (46.9%) had visited a doctor, 14 (43.8%) had not visited a doctor but considered a visitation, and 3 (9.4%) had not visited or considered consulting with a doctor (*p* < 0.001) (Table 3).

When the entire sample (*n* = 303) was tested, the mean endometriosis knowledge score value for the pre-test was 4.2 (SD = 2.48; SEM = 0.14), whereas it was 6.3 (SD = 2.3; SEM = 0.13) for the post-test (*p* < 0.0001). The paired-sample t-test yielded a mean EKS difference of 2.1 (95% CI = 1.88, 2.38), t = 16.63, df = 302, *p* < 0.001.

A one-way ANOVA was computed for the mean difference in endometriosis knowledge scores for the group with access to annual healthcare (AH = 150) and the group with inconsistent access to annual healthcare (IH = 153). Levene’s test of equality of the error of the variances yielded that the error of variance of the dependent variable was equal for both groups (*p* = 0.225). The difference between the groups with access to annual healthcare and inconsistent access to annual healthcare was statistically insignificant (*p* = 0.867). While the educational intervention seemed to have visually improved the endometriosis knowledge scores more for the AH group compared to the IH group, this finding was statistically insignificant (Figure 3).

## 4. Discussion

To the best of our knowledge, this study is the first screening assessment for the knowledge of endometriosis in Pakistan, which is representative of the general South Asian population. We developed the EKS, a specific questionnaire focusing on the origin, symptoms, and effects on fertility associated with endometriosis. Our findings demonstrated a significant concern among two-thirds of the women regarding their menstruation, and 10.6% suspected endometriosis as a potential cause for their worrisome symptoms. Of the women that attributed characteristics of the menstrual cycle with a probable diagnosis of endometriosis, less than half sought professional insight. We posited a knowledge gap among women regarding endometriosis based on the previous literature [22]. Similarly, we hypothesized that comprehensive education and insight into potential symptoms are typically provided by healthcare providers and are considered a prominent improvement in knowledge among women, regardless of their access to annual healthcare. Women who had consistent access to healthcare (Figure 3) had better knowledge of endometriosis, which pointed towards two key facilitators of improved quality of care: education and access to healthcare.

In the comparison of the group with constant access to annual healthcare and the group with inconsistent access to annual healthcare, the findings did not demonstrate significant improvements in endometriosis knowledge, despite the paired-sample t-test yielding a mean score increase of 2.1 (*p* < 0.001) across the entire sample after the intervention. This could be due to the normalization of menstrual symptoms by healthcare providers and patients [23,24,25]. Common symptoms of endometriosis are typically not further evaluated by healthcare physicians, suggesting clinical gender bias and societal stigma surrounding menstruation [26]. Healthcare providers have also been reported to have insufficient knowledge of potential symptoms of endometriosis, which requires further investigation. Another reason for the insignificant difference in the knowledge of endometriosis between women with and without healthcare access may be due to a lack of awareness, negative attitudes, and resistance to the recognition of the symptoms of endometriosis [27].

As reported previously, the gap in patients’ knowledge of endometriosis has been cited as one of the potential causes of delayed diagnosis [28]. We tested the baseline knowledge of the participants and identified a significant improvement in their understanding of the condition while focusing on improved symptom recognition and about the concept of subfertility. There is evidence of underestimated prevalence rates, particularly during adolescence and young adulthood, due to various factors, including education on endometriosis [29,30]. To the best of our knowledge, our study is the first to include information regarding menstruation to promote better awareness of the condition among women of reproductive age.

Other studies have explored the knowledge of endometriosis in different populations. A study conducted by Szymańska et al. in Poland found that 84% (*n* = 200) of women had heard of endometriosis, whereas only 4.5% of them had very good knowledge [31]. Another study reported by Zanden et al. in the Netherlands found that general practitioners (GPs) were not able to increasingly recognize endometriosis among patients, which lead to diagnostic delays [32]. It is pertinent to consider setting up teaching programs and strategies for GPs as first-line healthcare professionals in order to improve the detection rates and support timely diagnosis, as not all patients are directly referred to gynecologists.

It is critical to highlight that, beyond the lack of knowledge and awareness of endometriosis, there is immense societal stigma related to menstrual issues [33]. Further, there are no gold-standard noninvasive diagnostic tests for endometriosis [34]. The current treatment modalities focus on suppressing or removing endometrial lesions, but there is no clear understanding of the underlying contributory mechanisms [35]. Endometriosis is a chronic condition that poses many problems, including chronic pain and infertility, which, consequently, impact the quality of life of the patients [36]. Women who are diagnosed with endometriosis require comprehensive yet individualistic management of their conditions with a focus on their quality of life and fertility problems [35]. The rationale for treatment is often guided by the patient’s needs; the treatment may be medical or surgical, but a conservative approach is preferred unless the severity of symptoms is not controlled or medical treatment has failed [37]. Surgical management of endometriosis is subject to the patient’s desire for future pregnancies, the severity of the symptoms and disease, and the location of the disease [38].

### Limitations

The self-reported nature of our study has certain advantages and limitations. Given the sensitive nature of the information, it may be expected that the responses were more accurate and frequent than when collected by interviewers. We expect there to be some degree of recall bias, as the participants were asked to recall symptoms from 3 months before the survey. We could ascertain the intra- and inter-group differences in demographics and baseline characteristics beyond age, level of education, healthcare worker status, and access to healthcare. Regardless, as the EKS measurement was repeated, these biases are not expected to impact the improvement in the educational understanding of endometriosis. A link to the study was disseminated online in Pakistan, and the applicability of the findings may primarily be in this geographical region. The current study also did not incorporate women who were Internet-illiterate. As the study also included healthcare workers, the improvement in the knowledge of endometriosis may have been underestimated.

## 5. Conclusions

Our study incorporated a specific educational intervention for improving the knowledge regarding endometriosis among women. It is critical to focus on educational strategies among physicians and menstruating women to address diagnostic delay and resistance, as well as to bridge the knowledge gaps concerning endometriosis. This study highlights the need for health literacy, particularly concerning menstrual symptoms among young women. A collaborative effort by primary care providers, gynecologists, and educational systems is required to incorporate comprehensive screening and diagnostic strategies for women with endometriosis.

## Figures and Tables

**Figure 1 healthcare-11-00809-f001:**
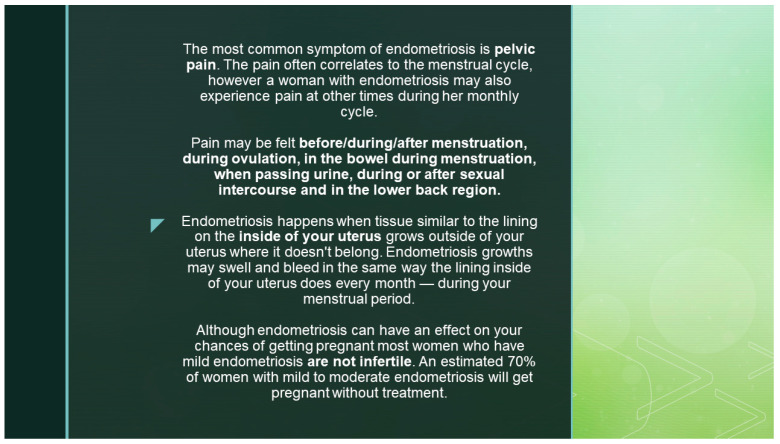
Educational brochure on endometriosis.

**Figure 2 healthcare-11-00809-f002:**
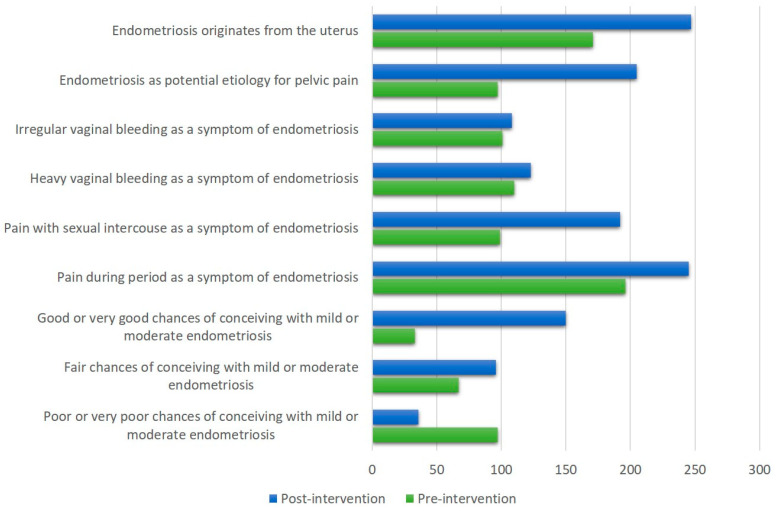
Participants’ responses to questions on endometriosis knowledge pre-and post-intervention.

**Figure 3 healthcare-11-00809-f003:**
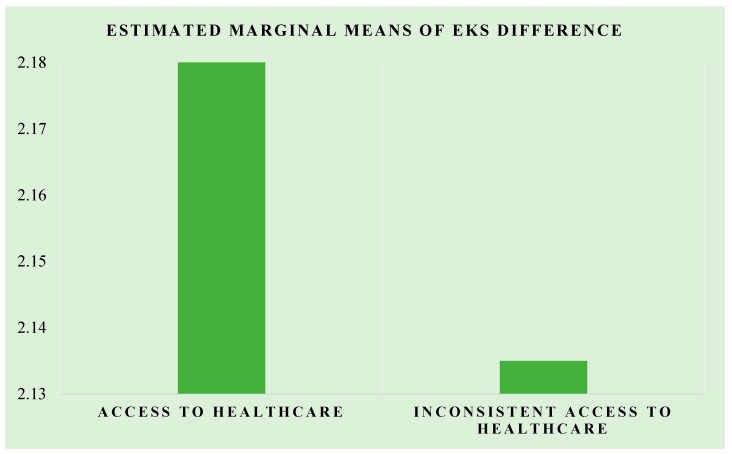
The plot of the estimated marginal means of the difference for the AH and IH groups.

**Table 1 healthcare-11-00809-t001:** The ten-point endometriosis knowledge score (EKS).

Question	Responses
Where does endometriosis originate from? (1 point)	Uterus—1 Fallopian tubes—0 Ovaries—0 Vagina—0 Do not know—0
What are potential etiologies for pelvic pain? (3 points)	Endometriosis—2 Any other—1 Do not know—0
Which of the following are symptoms of endometriosis? (4 points)	Irregular vaginal bleeding—1 Heavy vaginal bleeding—1 Pain with sexual intercourse—1 Pain during periods—1 Do not know—0
What are the chances that a woman with mild or moderate endometriosis will conceive? (2 points)	Very good/good—2 Fair—1 Poor/very poor/do not know/maybe—0

**Table 2 healthcare-11-00809-t002:** Sample characteristics (*n* = 303).

Characteristic	Number (%); *n* = 303	*p*-Value
Age		*p* = 0.405
<18 years	14 (4.6)	
18–29 years	236 (77.9)	
30–39 years	39 (12.9)	
>40 years	14 (4.6)	
Level of education		*p* = 0.993
Middle/Intermediate school	30 (9.9)	
High school	27 (8.9)	
Undergraduate	137 (45.2)	
Graduate	109 (36)	
Healthcare worker status		*p* < 0.001
Yes	76 (25.1)	
No	227 (74.9)	
Access to annual healthcare		*p* = 0.867
Always	150 (49.5)	
Sometimes	55 (18.2)	
Often	54 (17.8)	
Rarely	23 (7.6)	
Never	21 (6.9)	
Age at menarche		*p* = 0.905
<10 years	11 (3.6)	
10–16 years	287 (94.7)	
>16 years	5 (1.7)	
Duration of cycle		*p* = 0.856
<21 days	8 (2.6)	
22–24 days	31 (10.2)	
25–28 days	118 (38.9)	
29–32 days	92 (30.4)	
33–35 days	19 (6.3)	
>36 days	2 (0.7)	
Irregular	33 (10.9)	

**Table 3 healthcare-11-00809-t003:** Pre-intervention satisfaction with knowledge of symptoms, symptoms observed, and level of concern of the entire population (*n* = 303).

Characteristic	Number (%); n = 303	*p* Value *
Satisfaction with the knowledge of symptoms related to periods		*p* = 0.008
Completely satisfied	32 (10.6)	
Somewhat satisfied	117 (38.6)	
Neither satisfied nor dissatisfied	73 (24.1)	
Somewhat dissatisfied	57 (18.8)	
Completely dissatisfied	24 (7.9)	
Pelvic pain with periods in the last 3 months		*p* = 0.126
Always (3 of 3 periods)	88 (29)	
Often (2 of 3 periods)	68 (22.4)	
Occasionally (1 of 3 periods)	92 (30.4)	
Never	55 (18.2)	
Pelvic pain in between one period cycle and the next in the last 3 months		*p* = 0.903
Always (3 of 3 periods)	24 (7.9)	
Often (2 of 3 periods)	41 (13.5)	
Occasionally (1 of 3 periods)	95 (31.4)	
Never	143 (47.2)	
Pelvic pain while urinating or defecating during periods in the last 3 months		*p* = 0.02
Always (3 of 3 periods)	21 (6.9)	
Often (2 of 3 periods)	22 (7.3)	
Occasionally (1 of 3 periods)	66 (21.8)	
Never	194 (64)	
Concern about any symptoms related to periods		*p* < 0.001
Extremely concerned	40 (13.2)	
Moderately concerned	47 (15.5)	
Not at all	92 (30.4)	
Slightly concerned	72 (23.8)	
Somewhat concerned	52 (17.2)	
Suspected that symptoms related to periods are due to endometriosis		*p* < 0.001
No, and did not consider	209 (69)	
No, but considered	62 (20.5)	
Yes	32 (10.6)	

* Levels of associations were made with AH and IH as the key variates, with all other variables as covariates.

## Data Availability

All data presented in this study are presented within the manuscript. For any queries, the dataset may be obtained from the corresponding author (A.S.) upon reasonable request.

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
