# Peer review of "Addressing the Endometriosis Knowledge Gap for Improved Clinical Care—A Cross-Sectional Pre- and Post-Educational-Intervention Study among Pakistani Women"

_healthcare, 2023, doi:10.3390/healthcare11060809_

Round 1
Reviewer 1 Report
The article refers to a relevant problem – and offers useful information of a specific population.
Ther are some minor comments:
1. I think that there is a too descriptive development of the endometriosis stages (lines 34-40). A cited classification would suffice.
2. It is not clear “Symptoms experienced by women with endometriosis include pelvic pain during menstruation associated with urination and defecation“ (lines 42-43). Pelvic pain during menstruation is not necessarily associated with urination and defecation.
3. AH and AI acronyms appear first without explanation (line 93) – and this is solved only several lines below (lines 96-97).
4. The information in lines 124-146 is repeated in Table 3. One way should be chosen.
5. What significance refers to the P-values (lines 127, 136, 139, 146) – the ones in Table 3?
6. Figure 3 might not be appropriate – as it shows a link in-between the value of the respective groups – and it might suggest a progressive relationship. Columns would pe preferable.
7. I do not understand “However, the limitation is reporting and recall bias.” (lines 211-212).
Some major concerns could be related, especially to the difference bias in-between the groups (demographics, symptoms, education, etc.) and to the reported assessed differences:
1. Group 2 is rather heterogenous: “women with inconsistent access i.e., often/sometimes/rarely/never have access to annual healthcare”. More, if we consider the “often” subgroup, it would be difficult to see the difference from group 1. Conversely, including in group 2 only the women with rarely/never access to annual healthcare could unveil a difference from group 1.
2. Table 3 has no legend. It is not clear which were the respective characteristics of the two groups. Maybe the p > 0.05 express that there was not a significant difference in-between the groups – but this is not clear.
3. The authors include notions as “societal stigma” “normalization of menstrual symptoms by healthcare providers and patients”. It is not clear if these concepts are related to the study, as they were not tackled. More, they write that their “findings demonstrate a significant concern among 2/3rd of the women regarding their menstruation” (lines 170-171), which seems to be in contradiction.
Author Response
To the Esteemed Reviewer, I thank you very much for your time and attention in reviewing our very important paper. Your voluntary efforts are deeply and greatly appreciated. With Kind Regards,
Zouina S.
Reviewer 1 Comments and Author Responses:
The article refers to a relevant problem – and offers useful information of a specific population. There are some minor comments:
Reviewer 1, Comment 1: I think that there is a too descriptive development of the endometriosis stages (lines 34-40). A cited classification would suffice.
Author Response: While not the focus of the paper, we think it is beneficial to readers to understand the different stages of endometriosis as the condition itself is not clearly understood, even by healthcare professionals.
Reviewer 1, Comment 2: It is not clear “Symptoms experienced by women with endometriosis include pelvic pain during menstruation associated with urination and defecation” (lines 42-43). Pelvic pain during menstruation is not necessarily associated with urination and defecation.
Author Response: That is correct that pelvic pain or discomfort is not always associated with pain during urination and/or defecation. However, pelvic pain with urination/defecation is a well-recognized symptom of endometriosis throughout literature – attaching the information available by the World Health Organization https://www.who.int/news-room/fact-sheets/detail/endometriosis
Reviewer 1, Comment 3: AH and AI acronyms appear first without explanation (line 93) – and this is solved only several lines below (lines 96-97).
Author Response: The full forms of AH and IH have been added on line 93 to avoid any confusion.
Reviewer 1, Comment 4: The information in lines 124-146 is repeated in Table 3. One way should be chosen.
Author Response: The sentence seemed redundant and has been re-written to emphasize that majority of the sample was satisfied as it was a significant association. We have retained only a part of the information that seems relevant for the readers.
Reviewer 1, Comment 5: What significance refers to the P-values (lines 127, 136, 139, 146) – the ones in Table 3?
Author Response: The p-values were computed to understand the baseline differences of knowledge of symptoms, symptoms observed, and level of concern among participants.
I have added this in the text for more clarity:
“Across the entire cohort of women (N=303), the independent variables (AH=Always have access to healthcare; IH; Inconsistent access to healthcare) were associated to all other variables where both inferential/descriptive statistical findings were tabulated in Table 3.”
Reviewer 1, Comment 6: Figure 3 might not be appropriate – as it shows a link in-between the value of the respective groups – and it might suggest a progressive relationship. Columns would be preferable.
Author Response: We agree that the figure itself may suggest a gradient and it has been changed in favor of a bar graph.
Reviewer 1, Comment 7: I do not understand “However, the limitation is reporting and recall bias.” (lines 211-212). Some major concerns could be related, especially to the difference bias in-between the groups (demographics, symptoms, education, etc.) and to the reported assessed differences:
Author Response: Recall bias occurs when participants may not be able to completely recall past events as the format of the study was self-administered survey. The sentence has been re-written to elaborate and make it clear what we found was the bias.
Reviewer 1, Comment 8: Group 2 is rather heterogenous: “women with inconsistent access i.e., often/sometimes/rarely/never have access to annual healthcare”. More, if we consider the “often” subgroup, it would be difficult to see the difference from group 1. Conversely, including in group 2 only the women with rarely/never access to annual healthcare could unveil a difference from group 1.
Author Response: While we agree that the “often” subgroup entails of women who may have higher access to healthcare services than those in the “rarely/never” subgroup, we think the “always” subgroup could be analyzed separately. This is because these women perceived their access as all-encompassing whereas the “often” and “rarely” may be subjective to the respondent’s perception.
Reviewer 1, Comment 9: Table 3 has no legend. It is not clear which were the respective characteristics of the two groups. Maybe the p > 0.05 express that there was not a significant difference in-between the groups – but this is not clear.
Author Response: Table 3 represents findings of the entire cohort of women in our cross-sectional study. I have added a * below, which indicates that the levels of associations were made between IH and AH groups for better complacence and understanding.
“Levels of associations were made with AH and IH as the key variate with all other variables as covariates.”
Reviewer 1, Comment 10: The authors include notions as “societal stigma” “normalization of menstrual symptoms by healthcare providers and patients”. It is not clear if these concepts are related to the study, as they were not tackled. More, they write that their “findings demonstrate a significant concern among 2/3rd of the women regarding their menstruation” (lines 170-171), which seems to be in contradiction.
Author Response: We focus on the challenges and barriers for women who need to be evaluated for endometriosis. These concepts set the ground for what women face – including taboo and normalization. These are, however, introduced to elaborate on the diagnostic challenge of endometriosis only.
Regarding the second comment, if we check the response for “Concern about any symptoms related to periods” in table 3, nearly 69% of the women show some concern about their menstruation which is what we wrote in the discussion.
Reviewer 2 Report
The study was conducted amongst a low income cohort of women with limited access to good quality heathcare.
The main question here is how can we improve the overall quality fo care. Education is one aspect of this, but access to healthcare is a more important intervention. I would like to see how the Education was matched with better access, or at least, how the authors are considering that.
A more targeted reflection on interventions needed would be needed.
Author Response
To the Esteemed Reviewer, I thank you very much for your time and attention in reviewing our very important paper. Your voluntary efforts are deeply and greatly appreciated. With Kind Regards,
Zouina S.
Reviewer 2 Comments and Author Response:
The study was conducted amongst a low-income cohort of women with limited access to good quality healthcare.
The main question here is how can we improve the overall quality of care. Education is one aspect of this, but access to healthcare is a more important intervention. I would like to see how the Education was matched with better access, or at least, how the authors are considering that.
A more targeted reflection on interventions would be needed.
Author Response: We differentiated groups based on consistent and inconsistent access whereas education was provided to all women. We expected that women who have regular access to healthcare must have higher knowledge if given education on their symptoms. We have added relevant lines to emphasize the importance of both education and access in lines 178-180.
Reviewer 3 Report
I read with great interest the manuscript, which falls within the aim of this Journal. In my honest opinion, the topic is interesting enough to attract the readers’ attention. Nevertheless, authors should clarify some points and improve the introduction. Authors should consider the following recommendations: - Manuscript should be further revised in order to correct some typos and improve style. - It would appreciable a small topic on obstetrics outcomes and its role on QoL of women with endometriosis referring to PMID 34623489 and doi 10.1186/s10397-021-01096-5 - Moreover another topic on the role of surgical and medical intervention on endometriosis and DE referring to PMID: 34008386 and PMID: 33687160Author Response
To the Esteemed Reviewer, I thank you very much for your time and attention in reviewing our very important paper. Your voluntary efforts are deeply and greatly appreciated. With Kind Regards,
Zouina S.
Reviewer 3 Comments and Author Responses:
I read with great interest the manuscript, which falls within the aim of this Journal. In my honest opinion, the topic is interesting enough to attract the readers’ attention. Nevertheless, authors should clarify some points and improve the introduction. Authors should consider the following recommendations:
- Manuscript should be further revised in order to correct some typos and improve style.
- It would appreciable a small topic on obstetrics outcomes and its role on QoL of women with endometriosis referring to PMID 34623489 and doi 10.1186/s10397-021-01096-5
- Moreover, another topic on the role of surgical and medical intervention on endometriosis and DE referring to PMID: 34008386 and PMID: 33687160
Author Response: We have reviewed the manuscript for grammar and vocabulary. We have also added PMID 34623489, 34008386 and 33687160 in the discussion. Thank you for the great suggestion.
Reviewer 4 Report
Thank you very much for the invitation to review of the manuscript. It a great pleasure for me.
The purpose of Saad et al. was to explore the baseline knowledge of endometriosis background and symptoms associated with the disease, and documented the improvement in endometriosis knowledge. This is very interesting paper, However, I have a few suggestions and advice:
1. In the introduction, you can add information about the general knowledge of women about endometriosis
2. It seems that the group of respondents should be better characterized, what was the material status, level of education in the group over 18 ... Did the surveyed women use contraception or give birth?
3. Did these women have before any complaints from the reproductive system, which could increase their awareness of the disease?
4. In the discussion chapter, it is worth adding the results of observations from other countries.
Author Response
To the Esteemed Reviewer, I thank you very much for your time and attention in reviewing our very important paper. Your voluntary efforts are deeply and greatly appreciated. With Kind Regards,
Zouina S.
Reviewer 4 Comments and Author Responses:
Thank you very much for the invitation to review of the manuscript. It a great pleasure for me.
The purpose of Saad et al. was to explore the baseline knowledge of endometriosis background and symptoms associated with the disease, and documented the improvement in endometriosis knowledge. This is very interesting paper; However, I have a few suggestions and advice:
Reviewer 4, Comment 1: In the introduction, you can add information about the general knowledge of women about endometriosis
Author Response: Studies that we found on knowledge of women and healthcare professionals have been added in the discussion on lines 202-209.
Reviewer 4, Comment 2: It seems that the group of respondents should be better characterized, what was the material status, level of education in the group over 18 ... Did the surveyed women use contraception or give birth?
Author Response: Thank you for the feedback. We did not ask of all these variables yet we inquired on age, level of education, healthcare worker status, age at menarche, duration of cycle, and access to healthcare.
Reviewer 4, Comment 3: Did these women have before any complaints from the reproductive system, which could increase their awareness of the disease?
Author Response: We asked separately regarding their awareness of endometriosis including if they had heard of it. By doing so, we accounted for pre-intervention awareness of disease.
Reviewer 4, Comment 4: In the discussion chapter, it is worth adding the results of observations from other countries.
Author Response: We have added studies from other countries on lines 202-209.
Reviewer 5 Report
The paper itself is well written, but it should be added:
1.. Add "pilot study" to the title
2. Discussion compared with the introduction is too short.
3. The number of the bioethics committee should be provided.
Author Response
To the Esteemed Reviewer, I thank you very much for your time and attention in reviewing our very important paper. Your voluntary efforts are deeply and greatly appreciated. With Kind Regards,
Zouina S.
Reviewer 5:
The paper itself is well written, but it should be added:
Reviewer 5, Comment 1: Add "pilot study" to the title
Author Response: The term “cross-sectional” study has been added to the title. Upon discussing with the committee and colleagues, the term “pilot” is rendered an ‘un-original’ study in our department/region. Hence it is requested to accept the alternative. I thank you for your time and attention to this matter.
Reviewer 5, Comment 2: Discussion compared with the introduction is too short.
Author Response: The discussion has been improved and more literature has been added.
Reviewer 5, Comment 3: The number of the bioethics committee should be provided.
Author Response: The information of the bioethics committee is linked to this drive folder: Please have a look:
https://drive.google.com/file/d/1fvHfv8vez75pakLHIac--3f7COarEI8m/view?usp=sharing.
Round 2
Reviewer 1 Report
NA
Reviewer 5 Report
Thank you for your revised version.